# Sexual Violence and Substance Use Diagnosed in Emergency Department Discharges in Hawaii, 2005–2014

**DOI:** 10.3390/ijerph192316220

**Published:** 2022-12-04

**Authors:** Mary Guo, Bobby Do, Korben Wong, Thomas H. Lee, Deveraux Talagi, Brandon Lum, Nichole Rahberg, Edra Ha, Victoria Y. Fan

**Affiliations:** 1Pacific Health Analytics Collaborative, Social Science Research Institute, College of Social Sciences, University of Hawaii at Manoa, Honolulu, HI 96822, USA; 2Office of Planning, Policy and Program Development, Department of Health, Honolulu, HI 96813, USA

**Keywords:** sexual violence, domestic violence, substance use, emergency medicine, hospital services

## Abstract

Background: Substance use (SU) and sexual violence (SV) present unique challenges when contextualizing their relationship due to underreporting of SU and SV. Both are significant public health concerns with a large magnitude and expense to the overall U.S. and to the state of Hawaii, which is identified as a high-intensity drug-trafficking area. Since substance users have a higher risk for sexual violence than the general public, this study aims to analyze the proportion and demographics of emergency department (ED) visit individuals reporting sexual violence with or without substance use disorder and examine how the number of ED visits of individuals diagnosed with SV and SU disorder compare to all individuals. Methods: Data from the Healthcare Cost and Utilization Project was used to examine the relationship between SV and SU. The database contained 3.5 million observations for 24 Hawaii hospitals from 2005–2014. The data was summarized in descriptive statistics and Chi-square tests were run to assess statistical significance for variables of interest. Results: A greater proportion of individuals reporting sexual violence also reported substance use disorders compared to the general population of individuals. While 8% of all ED visits were related to SU, 17% of ED visits involving SV were also related to SU, demonstrating a statistically significant association between SV and SU. Conclusions: There is a greater need to further understand the complexity of the relationship between substance use and sexual violence. Sexual violence and substance use disorders share a complex relationship; survivors of sexual abuse may develop a substance use disorder, and those who use drugs and alcohol may be at an increased risk for sexual violence. Results from this study demonstrate visits for individuals reporting sexual violence have a greater proportion of substance use disorder than visits for the general population.

## 1. Introduction

Substance use (SU) and sexual violence (SV) are significant public health and medical concerns. SU includes use of alcohol, tobacco, and illicit drugs, such as heroin, cocaine, marijuana, and methamphetamine [1]. In 2020, 40.3 million people aged 12 or older in the U.S. had a substance use disorder in the past year, including 28.3 million and 18.4 million with an alcohol use disorder and illicit drug use disorder, respectively [1].

The effects of SU are broad and wide-ranging, resulting in a variety of public health and public safety issues, including motor vehicle crashes, crime, violence, and child abuse [2]. Nationally, SU amounts to over $740 billion annually in costs related to crime, lost work productivity, and health care [3].

In particular, sexual violence (SV) is sexual activity when consent is not given. According to the National Center for Injury Prevention and Control, Division of Violence Prevention, SV affects millions of people each year in the United States and is often perpetrated by someone known by the survivor or victim [2,4]. Yet, SV is challenging to study due to lack of reporting. Many cases of SV are unreported because victims may be ashamed, embarrassed, or afraid to tell anyone about the crime, and estimates indicate one in every three women and one in every four men have likely experienced sexual violence involving physical contact during their lifetimes [5,6]. According to a Justice Department analysis of violent crime in 2016, nearly 80 percent of rapes and sexual assaults go unreported [7]. The cost of rape is estimated at $122,461 per victim including medical costs, lost productivity, criminal justice activities, and other costs [5].

The focus of this study is the association between SV and SU, which has been studied and documented. Perpetrators of SV often use alcohol and drugs as tools to facilitate an assault by incapacitating and discrediting victims [8]. Because alcohol and substance use may be coping mechanisms for psychological traumas in general, alcohol and substance use may also be a consequence of SV and not merely a risk factor of consequence of SV [9]. In short, the causal relationship between SV and SU are not well identified.

In the state of Hawaii, the focus of this study, the last publicly available report on SV in Hawaii was published over 15 years ago by the Sex Abuse Treatment Center (SATC) and Department of the Attorney General; the report provided a descriptive analysis of over 5000 sexual assault victims in a 12-year period who received treatment or services through the SATC in Honolulu, Hawaii [10]. Hawaii is also classified as a “high-intensity drug trafficking area” (HIDTA), where ice methamphetamine, marijuana, controlled prescription drugs, cocaine, and heroin use are the leading drug concerns [11]. In Hawaii, the age-adjusted number of deaths due to drug injury was 12.3 per 100,000 population [12]. The annual cost of underage drinking in Hawaii totals $200,000 million, which includes tangible costs of $106.3 million (including medical care, criminal justice, property damage, and loss of work) and pain and suffering costs of $113.2 million [13]. Finally, the annual costs of smoking-related health care and smoking-related losses in productivity in Hawaii are estimated at $526 million and $387.3 million, respectively [14]. These numbers indicate SU is a significant public health concern with a large magnitude and expense to the nation overall and to the state of Hawaii specifically.

The overarching purpose of this study is to better examine the relationship between SV and SU using an important data resource with potential for screening of SV and SU. This study focused on the intersection of SV and SU to shed greater light on two important safety and health challenges. The study examined two primary questions. First, how do the proportion and demographics of individuals who visit the ED reporting sexual violence with or without substance use disorder compare to the general population of individuals? Exploration of this first question is essential to the identification of the target population for interventions and recommendations for health policy changes.

Second, how do the number of ED visits of individuals diagnosed with SV and SU disorder compare to the general population of individuals visiting the ED? The ED is a point of entry for patients experiencing sexual assault and plays a critical role by providing immediate care and connecting patients with community resources for sexual violence survivors [15]. This makes ED data a valuable resource for collecting information on characteristics associated with SU and SV. The study sought to identify a target population for potential health policy changes or prevention strategies from a demographic perspective, i.e., with a focus on individuals experiencing both SV and SU.

## 2. Methods

### 2.1. Comprehensive Emergency Department Visit Data Resource

The data used for this study are a comprehensive, population-level resource of all outpatient emergency department (ED) visits in the state of Hawaii, covering all outpatient ED visits in the state. The data are obtained from the State Emergency Department Databases (SEDD), which is part of the Healthcare Cost and Utilization Project (HCUP), sponsored by the Agency of Healthcare Research and Quality. HCUP is the largest collection of longitudinal hospital care data in the United States. The database is dependent upon state organizations to submit their data voluntarily [16]. In the state of Hawaii, HCUP and SEDD data are available only through 2014, after which the hospital association stopped reporting data to the HCUP.

This database is valuable because it is comprehensive of all emergency department visits in the state of Hawaii and is a universe of all visits; it is not a sample but the full population. Data on visits to all 24 hospitals with emergency departments in the state of Hawaii from 2005 to 2014, totaling 3.5 million observations, were summarized in descriptive statistics to illuminate demographics and trends of the relationship between SU and SV in recent years. The unit of analysis is the ED encounter, not the individual patient. This means that a person who is seen in the ED multiple times per year is counted each time in the dataset as a separate “encounter” in the ED.

### 2.2. Key Variables of Interest

Variables, including substance use and sexual violence-related International Classification of Diseases, Ninth Revision (ICD-9) codes, gender, race, and age, were explored. ICD-9 assigns numeric codes to diagnoses. SV-related visits were defined as having an ICD–9 code of 99,583 (sexual abuse adult) or 99,553 (sexual abuse child). The type of substances involved are indicated by a list of 67 ICD–9 codes. Refer to Table 1 for the list of ICD-9 codes used to identify SV- and SU-related visits.

Gender was reported as female or male. Race was reported as White/Hispanic, East Asian, Other Asian, Pacific Islander, or Other. Note that from 2010 to 2014, HCUP made a recategorization of race groupings, thus causing slight variations and limiting subcategorizations of race. Refer to Table 1 for a list of definitions for the race values. In addition, inconsistencies with the methodology for race categorization in 2010 led to visits from 2010 having to be omitted for racial statistics.

### 2.3. Software, Cell Suppression, and Ethics Review

R software (Version 3.6.0) (The R Foundation, 2019) was used to conduct data analyses. All details that might disclose participants’ identities are not reported. Chi-square tests were run to assess for statistical significance in relationships between variables of interest. This study was approved by the University of Hawaii Institutional Review Board as exempt (2019-00573) because the data used historical de-identified secondary administrative data with no personal identifiers for the purpose of research and public health reporting.

Note, the HCUP data use agreement requires each table cell to have no fewer than 11 individuals per cell to maintain confidentiality; thus, some of the tables combined the columns for visits involving both SV and SU and the visits involving SV but not SU into an aggregate column for any visits involving SV to avoid cell suppression.

## 3. Results

Of the 1.7 million individuals with ED visits in Hawaii from 2005 to 2014, 153,788 ED-individuals made up 281,784 of ED visits related to SU and/or SV. Overall, visits related to SU and/or SV varied by substance use type and the patient’s sex, age group, and race. There were more than 3.5 million ED visits, indicative there were multiple ED visits by the same individuals over this period. As demonstrated in Table 2, the most common SU-related ED admissions were tobacco use disorder and alcohol abuse; tobacco use disorder and alcohol abuse accounted for 70% and 16%, respectively, of all SU-related ED visits. The third most common substance use was amphetamine abuse, accounting for nearly 6% of ED visits related to SU.

Table 3 shows ED visit counts related to SV were low, comprising less than 0.01% of all ED visits. While 8% of all ED visits were related to SU, 17% of ED visits involving SV were also related to SU. The Chi-square statistic of 14.2 shows a statistically significant association between SV and SU at *p* < 0.01.

Table 4 illustrates the variations in the number of ED visits made by individuals depending on if they had ED visits involving SV and/or SU. Overall, 1,677,090 ED-individuals made approximately 3.5 million visits to the ED during the study period. Out of that total, 153,788 individuals had 281,784 ED visits related to SU and/or SV. Out of all the ED visit individuals, 20% had 3+ ED visits from 2005 to 2014. In contrast, 55% of individuals with ED visits involving SV and 43.5% of individuals with ED visits involving SU but not SV had 3+ ED visits over the 10-year duration.

Table 5 shows nearly one in every ten ED visit by men were related to SU, while the proportion of SU for females was lower at approximately 6%. The proportion of ED visits related to SV was roughly eight times greater for females than males; 0.008% and 0.001% of ED visits for females and males, respectively, were related to SV, whether alone or in combination with SU. Approximately one in every ten ED visits for adults ages 20 to 49 were related to SU and not SV, and adults ages 40 to 49 had the greatest proportion of ED visits related to SU and not SV out of all the age categories at 13%. Individuals in their second and third decades of life (10–19 years and 20–29 years) had the greatest proportion of SV related ED visits compared to the other age categories at 0.011% and 0.009% of all their ED visits, respectively. The Other race group had the greatest proportion of ED visits related to SV at 0.008%, while the White/Hispanic race group had the greatest proportion of ED visits involving SU but not SV with roughly one in ten ED visits.

## 4. Discussion

The search for solutions to improve the nation’s health is ongoing, and two main objectives of public health intervention efforts are to decrease rates of substance use and sexual violence. This study investigates the relationship between SU and SV to identify a target population, as well as the frequency of ED visits of those admitted for SU and/or SV, for interventions and recommendations for health policy changes. The study emphasizes the importance of using hospital data, including emergency department visits for prioritizing prevention and/or screening of SV and SU.

Overall, the HCUP reported proportions of substance use are on par with national percentages [17], but the comparison of HCUP reported proportions of sexual violence to national rates is a challenge, as the vast majority of national sexual violence report rates are reported as lifetime rates [5,6,18]. As a result, the sexual violence rates from the HCUP data based on encounters over a 10-year span are lower than national lifetime percentages.

Over a ten-year period, there were 151 ED visits related to SV for all ages, which seems to be an underestimate of the number of SV cases. In comparison, there were over 5000 visits to the SATC in Hawaii over a twelve-year period. There are several plausible interpretations of this underestimate. First, some of the SU and SV victims may have been moved to the inpatient setting; these are excluded from our counts because SEDD comprises outpatient data. Second, survivors may not report their sexual assaults in a medical encounter, perhaps due to lack of training and familiarity by healthcare providers about how to diagnose and treat such conditions or due to fear by the survivor of retaliation by the perpetrator. Third, if SU is involved, then SV cases may appear to be an underestimate as substance use affects people’s cognition, emotions, and memory.

Whereas this study largely considers the number of ED visits, we also examined the number of unique individuals and the number of ED visits those individuals had. While synthetic medical record numbers allowed tracking of multiple visits by an individual to a given facility, they do not uniquely identify individuals across hospital facilities. As individuals who visited a given ED may have visited another ED, the overall number of individuals is an overestimate. In addition, the number of ED visits made by individuals is an underestimate since individuals who use more than one ED are captured by each ED as a unique individual. Other sources of ED data may be valuable to explore for additional analyses by unique individuals.

## 5. Limitations

A cross-study comparison of proportions from this study is a challenge due to variation among existing studies in defining, identifying, and subtyping cases of suspected drug-facilitated sexual assault [8]. In general, research is sparse on national hospital data, including ED visit data, which reviews substance use with sexual violence rates. The HCUP data shows, in general, women (versus men) and those in their second and third decades of life (versus all other ages) are at highest risk of sexual violence. These findings are consistent with a recent study on young adult opioid users’ sexual experiences in the context of their drug use, which found sexual assault was a frequent occurrence, such as when the victims were unconscious or were forced to have sex in exchange for drugs or money [19].

Despite the challenges in drawing a cross-study comparison of findings and limitations in data interpretation, this study was valuable for multiple reasons. The HCUP data was helpful in beginning to describe the gender and age group distributions of SU- and SV-related ED visits in recent years. For example, the data showed a greater number of males than females were diagnosed with a substance use code, while more females than males were diagnosed with a sexual assault code. Future studies should review additional variables, such as ED visits by geography, marital status, and insurance type (private versus public), to better understand the demographics of the population of frequent users of the ED. Future studies should also consider differentiating between offender substance use and survivor substance use and its impact on sexual assault. However, due to the cell size suppression rules in accordance with the data use agreement and the small population measured in this study, reporting on all the disaggregated characteristics remains challenging.

Furthermore, this research supports the Hawaii Opioid Initiative launched in July 2017 by Governor David Ige in response to the national opioid crisis to develop and implement a proactive coordinated statewide action plan [20]. In particular, further investigation of the relationship between SU and SV is an opportunity to organize and encourage coordination among stakeholders to address substance use issues in collaboration with teams which target sexual violence reduction. As an example, the relationship between sexual assault and alcohol use could underscore primary prevention efforts in multiple at-risk populations.

As the data used for this study are dated, it neglects the potential study of new psychoactive substances (NPS) which may be contributing to the growth in substance use seen across the United States including the state of Hawaii [21,22]. New substances are also harder to diagnose, including difficulty to detect in laboratory samples, let alone in hospital data, which are not necessarily linked to laboratory data.

## 6. Conclusions

Sexual violence and substance use disorders share a complex relationship; survivors of sexual abuse may develop a substance use disorder, and those who use drugs and alcohol may be at an increased risk for sexual violence. Results from this study demonstrate ED visits for individuals reporting sexual violence have a greater proportion of substance use disorder than ED visits for the general population. Furthermore, individuals diagnosed with sexual violence and/or substance use disorder are more likely to have had three or more ED visits over the 10-year duration compared to the general population of diagnosed individuals. ED data is rich with valuable information and deserves more attention. This study utilized ED data to shed light on multiple opportunities for further investigations on the complex relationship between SU and SV.

These findings underscore the importance of further efforts to understand the complexity of the relationship between substance use and sexual violence to guide health policy and policy enforcement efforts. This study provides useful data for researchers and lawmakers to explore when they consider policy-driven interventions to prevent sexual violence and reduce substance misuse in the community.

## Figures and Tables

**Table 1 ijerph-19-16220-t001:** ICD-9 Codes for Sexual Violence and Substance Use: HCUP Emergency Department Visit Database for Hawaii, 2005–2014.

Category	Codes
Sexual Violence	
Sexual abuse adult	99,583
Sexual abuse child	99,553
Substance Use	
Alcohol Abuse	30,500–30,503
Alcohol Dependence	30,390–30,393
Alcohol Intoxication	30,300–30,303
Amphetamine Abuse	30,570–30,573
Amphetamine Dependence	30,440–30,443
Antidepressant Abuse	30,580–30,583
Barbiturate Abuse	30,540–30,543
Barbiturate Dependence	30,410–30,413
Cannabis Abuse	30,520–30,523
Cannabis Dependence	30,430–30,433
Cocaine Abuse	30,560–30,563
Cocaine Dependence	30,420–30,423
Combination of Drugs Dependence Excluding Opioid	30,480–30,483
Combination of Opioid Drug with Any Other Drug Dependence	30,470–30,473
Hallucinogen Abuse	30,530–30,533
Hallucinogen Dependence	30,450–30,453
Opioid Abuse	30,550–30,553
Opioid Dependence	30,400–30,403
Other Drug Abuse Mixed or Unspecified	30,590–30,593
Other Drug Abuse Specified	30,460–30,463
Other Drug Abuse Unspecified	30,490–30,493
Tobacco Abuse	30,510–30,513
Tobacco Use Disorder	3051

**Table 2 ijerph-19-16220-t002:** Visits for Substance Use Categories: HCUP Emergency Department Visits Database for Hawaii, 2005–2014.

Substance Use Type	Number of ED Visits
Alcohol Abuse	46,454
Alcohol Dependence	7872
Alcohol Intoxication	7699
Amphetamine Abuse	16,774
Amphetamine Dependence	1241
Antidepressant Abuse	36
Barbiturate Abuse	1005
Barbiturate Dependence	807
Cannabis Abuse	14,678
Cannabis Dependence	313
Cocaine Abuse	4330
Cocaine Dependence	246
Combination of Drugs Dependence Excluding Opioid	204
Combination of Opioid Drug with Any Other Drug Dependence	236
Hallucinogen	162
Opioid Abuse	3190
Opioid Dependence	4992
Tobacco Use Disorder	197,083
Other Drug Abuse Mixed or Unspecified	11,286
Other Drug Abuse Specified	306
Other Drug Abuse Unspecified	2088
Overall Any Substance Use	281,658

Note: Hallucinogen abuse and dependence were combined to maintain a cell size larger than 10.

**Table 3 ijerph-19-16220-t003:** ED Visits Involving Sexual Violence and/or Substance Use: HCUP Emergency Department Visit Database for Hawaii, 2005–2014.

Type of ED Visit	Number of Visits
SV but not SU	126
SU but not SV	281,633
Both SV and SU	25
Neither SU nor SV	3,270,737
Any ED Visit	3,552,521

Note: SV = sexual violence, SU = substance use.

**Table 4 ijerph-19-16220-t004:** Number of ED-Individuals with ED Visits Involving Sexual Violence and/or Substance Use in Hawaii, 2005–2014.

Type of ED Visit	Number of ED-Individuals with 1–2 Visits	Number of ED-Individuals with 3–4 Visits	Number of ED-Individuals with 5+ Visits	Number of ED-Individuals with Any Number of Visits
SV	66	25	56	147
SU but not SV	86,761	28,610	38,270	153,641
Neither SU nor SV	1,252,197	168,059	103,046	1,523,302
Any ED visit	1,339,024	196,694	141,372	1,677,090

Notes: SV = sexual violence, SU = substance use. HCUP data use agreement requires each table cell has no fewer than 11 individuals per cell. Hence, the number of ED visits was combined into the given categories. In addition, an ED-individual is not a unique individual but rather a unique individual for a given ED. These are not true unique individuals but rather unique individuals for a given hospital facility. The HCUP data provided do not uniquely identify individuals across hospital facilities. These numbers are therefore an underestimate of repeated ED visits since some individuals with only one visit for a given ED may have made another visit at another ED. Conversely, the overall number of individuals is an overestimate, i.e., the true number of unique individuals is less than what the table indicates.

**Table 5 ijerph-19-16220-t005:** ED Visits by Patient Sex, Age Group, and Race: HCUP Emergency Department Visits Database for Hawaii, 2005–2014.

	ED Visits Involving SV (%)	ED Visits Involving SU but Not SV (%)	ED Visits Involving Neither SU nor SV (%)	Total ED Visits
Overall	151 (0.004)	281,633 (7.93)	3,270,737 (92.07)	3,552,521
Sex				
Female	137 (0.008)	114,903 (6.36)	1,691,870 (93.63)	1,806,910
Male	14 (0.001)	166,728 (9.55)	1,578,830 (90.45)	1,745,572
Age Category				
0–9 Years	15 (0.003)	184 (0.033)	549,873 (99.96)	550,072
10–19 Years	39 (0.011)	14,588 (4.14)	338,184 (95.85)	352,811
20–29 Years	48 (0.009)	66,367 (11.76)	498,112 (88.24)	564,527
30–39 Years	21 (0.004)	57,899 (12.24)	415,302 (87.76)	473,222
40–49 Years	13 (0.003)	61,395 (13.12)	406,631 (86.88)	468,039
50+ Years	15 (0.001)	81,197 (7.10)	1,062,565 (92.90)	1,143,777
Race *				
White/Hispanic	57 (0.005)	103,490 (9.62)	972,431 (90.38)	1,075,978
Hawaiian/Pacific Islander	20 (0.003)	68,256 (8.56)	729,015 (91.44)	797,291
East Asian	11 (0.003)	20,202 (4.98)	385,433 (95.012	405,646
Other Asian	15 (0.003)	28,670 (5.56)	486,639 (94.43)	515,324
Other	23 (0.008)	25,496 (8.69)	268,012 (91.31)	293,531

Notes: SV = sexual violence, SU = substance use. HCUP data use agreement requires each table cell has no fewer than 11 individuals per cell. Due to this, the columns for ED visits involving SV but not SU and ED visits involving both SV and SU were combined into a single column for ED visits involving SV to avoid cell suppression. Age groups were reported for 10-year increments, and ages after age 50 were combined to maintain cell sizes of greater than 10. Pairwise deletion was used to exclude data with missing information for the sex, age, and race demographic numbers. * Due to inconsistencies in the race categorization methodology in 2010, ED visits from 2010 are omitted from the race counts.

## Data Availability

Please visit AHRQ HCUP website for more information: https://www.hcup-us.ahrq.gov/.

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
