# Peer review of "Sexual Violence and Substance Use Diagnosed in Emergency Department Discharges in Hawaii, 2005–2014"

_ijerph, 2022, doi:10.3390/ijerph192316220_

Round 1
Reviewer 1 Report
This paper presents epidemiologic data of the co-occurrence of substance use and sexual violence in emergency department encounters in Hawaii between 2005-2014. While the concept and question are important, I believe the manuscript needs to be improved in a number of key areas. First, the authors do not make a compelling case for the scale or impact of sexual violence in this database or study population. The actual N is quite small. Second, the only outcome that explores the impact of co-occurrence of these exposures is number of ED visits. Statistical analyses are minimal. Third, the dataset is limited to 2005-2014. It is unclear why more recent data are not utilized.
Introduction:
The authors state that sexual violence is a growing public health and medical concern but do not provide evidence to support this assertion. Sexual violence should also be defined (only rape?). The introduction also does not make a notable case about the impact of sexual violence on individuals, substance use or other outcomes, or society.
There are multiple associations between psychosocial and medical problems and substance use. However, less is known about the causal aspect of the relationship.
The references and estimates of prevalence of SU are 5 years or older and should be updated to within the last 2 years where more recent data is available
The term abuse is now considered stigmatizing and has been replaced by misuse. Additionally, person first language is now preferred. For example, substance users should be reworded something like individuals who use substances.
“Alcohol is the most commonly used substance in sexual assault and in turn, survivors may turn to alcohol and other SU as coping mechanisms for the trauma of SV and other associated social struggles.31” In tern implies a relationship between alcohol being the most commonly used substance in sexual assault and the fact that many victims self-medicate with alcohol. I do not believe this relationship is founded in evidence, though both may be true, true, and unrelated.
The introduction does not clearly set the stage for the study. The concluding two paragraphs could be clarified to define the central premise of the paper and the approach. The term “frequent flyers” is also stigmatizing and unnecessarily colloquial for a research paper.
Methods:
The design/management and availability of HCUP is unclear. It also was confusing as to what part of the dataset was utilized until later. It would be helpful to the reader to start with the subset of data utilized and then describe the larger database.
Are data not available since 2014? What is the reason for utilizing data that are now 8+ years old? It may be worth considering updating the dataset to be more reflective of current trends.
Use of subheadings for methods and results would greatly help with flow and anchoring for the reader.
Tables are not uniform and formatting is distracting. It would also be good to add concurrent ICD10 codes as that is the current coding system.
The age of patients is not clearly delineated.
Results
The term discharge seems to be incorrectly used. I’m assuming the authors are intending to look at encounter diagnosis or factors associated with the encounter, not the discharge.
Does the group involving SV include SU? Why not have SV only, SU only, SV + SU, and neither?
Reviewer 2 Report
This manuscript by M. Guo et al. consists of a cross-study about the relationship between sexual violence and substance use in the context of hawaiian emergency department, from 2005 to 2014, in order to provide characteristics of individuals discharged with diagnosis of sexual violence and substance use disorder compared to the general population of discharged individuals.
The manuscript is about a current and rising public health issue and it appears consistent with the recent literature. The aim is to identify a target population for health policy changes, trying to provide additional knowledge in the preventions strategies from a demographic perspective.
The current literature is widely focused on rising phenomenon of the substance use/misuse and the association with sexual violence or risky sexual behavior, involving themes like sexual minorities, chemsex, domestic violence, child sexual abuse and related suicide risk.
The introduction sets out the argument properly and illustrates the medical concerns and epidemiological data linked to the substance abuse in the United States with highlights related to Hawaiian context. The authors also cite social and political strategies and give information about economic direct and indirect costs related to the topic, giving a clear idea of the multi-dimensional impact of the phenomenon.
In the method section, demographic data and trends are collected from a valid, large and quite recent database, with the possibility of an appropriate sampling, and analyzed by a descriptive statistic. The variables are well presented, as well as the data use agreement. The study carries out sufficient data and gives clear explanation the methodology, specifying the meaning of unit of analysis (i.e. the emerging department encounter, not the individual patient).
In the result and discussion sections, the authors present data collection, showing understandable tables, referring to the categorization methodology and taking attention to the inconsistencies and to difficulties in data interpretation.
In the conclusions, the aims of the study seem to be achieved by the authors and the trends are plausible.
The list of references is enough recent and adequate.
In my opinion, there are few concerns that need to be clarified before considering the publication of the article.
In the abstract, the author should stress out the second aim of the study, in order to give a more complete synthesis of the work. Moreover, the keyword could be refilled, adding terms like “prevention policies”.
In the method section, ethical standards need to be clarified by the “World Medical Association Declaration of Helsinki” citation requested.
It is to notice a little possibility of misunderstanding about the number of individuals with emergency department discharges in Hawaii from 2005 to 2014 and the total discharges after observations in Hawaiian hospital in the same period (1.7 million vs 3.5 million).
In the discussion, would be appropriate referencing more published research to support the manuscript’s findings and emphasizes the originality of the study making a comparison with the emerging literature. Authors should also cite the essential impact on the drug scenario of the new psychoactive substances (NPS), taking information from studies like “New Psychoactive Substances and Suicidality: A Systematic Review of the Current Literature” (doi: 10.3390/medicina57060580) and “Focus on Over-the-Counter Drugs' Misuse: A Systematic Review on Antihistamines, Cough Medicines, and Decongestants” (doi: 10.3389/fpsyt.2021.657397). It is important to focus on the NPS emerging use to provide more updated data to the article. Moreover, the authors should illustrate the difficulties in detecting the NPS in the laboratory samples and consequently in the collecting data in hospitals databases, with the aim of comparing this concern with the result of the present study.
